# Neuropsychological Outcomes of Children Treated for Brain Tumors

**DOI:** 10.3390/children10030472

**Published:** 2023-02-27

**Authors:** Alessia Pancaldi, Marisa Pugliese, Camilla Migliozzi, Johanna Blom, Monica Cellini, Lorenzo Iughetti

**Affiliations:** 1Pediatric Oncology and Hematology Unit, Department of Medical and Surgical Sciences for Mother Children and Adults, University of Modena and Reggio Emilia, Via del Pozzo 71, 41124 Modena, Italy; 2Psychology Unit, Women’s and Children’s Health Department, University Hospital of Modena, 41124 Modena, Italy; 3Department of Biomedical, Metabolic and Neural Sciences, University of Modena and Reggio Emilia, 41124 Modena, Italy

**Keywords:** children, adolescents, CNS, brain tumor, neurologic late effects, radiotherapy, cognitive/neuropsychological outcomes

## Abstract

Central nervous system (CNS) neoplasms are the most common solid tumors diagnosed in children. CNS tumors represent the leading cause of cancer death and cancer-related morbidity for children less than 20 years of age, although there has been a moderate increase in survival rates over the past several decades. The average survival at 5 years now nearly reaches 75%, and for some, non-malignant histology approximates 97% at 20 years from diagnosis. Neurological, cognitive, and neuropsychological deficits are the most disabling long-term effects of brain tumors in children. Childhood is a time of extreme brain sensitivity and the time of life in which most brain development occurs. Thus, the long-term toxicities that children treated for CNS tumors experience can affect multiple developmental domains and day-to-day functioning, ultimately leading to a poor quality of survival (QoS). We reviewed literature focusing on the risk factors for cognitive and neuropsychological impairment in pediatric patients treated for brain tumors with the aim of better understanding who is at major risk and what the best strategies for monitoring these patients are.

## 1. Introduction

Neoplasms of the central nervous system (CNS) are the most common solid tumors diagnosed in children. Although there has been a moderate increase in survival rates in the last decades, CNS tumors still represent the main cause of cancer-related mortality and morbidity for children less than 20 years of age [1,2]. Five-year relative survival varies substantially by histological subtype. The relative survival at 5 years now approaches 75% for malignant tumors and 97.9% for non-malignant tumors, according to the data reported in the Central Brain Tumor Registry of the United States (CBTRUS) for the age range 0–19 years [1,3,4]. Significant survival discrepancy by ethnicity still exists for children with CNS tumors, with the largest disparities for diffuse astrocytoma and embryonal tumors [5]. The increase in survival also represents an increasing challenge for oncologists who have to minimize the late effects of tumor diagnosis and treatment. The term “late effects” is broad and applies to complications that begin and can persist after the tumor diagnosis, including lifelong complications. Survivors of childhood cancer can experience many late effects that may affect virtually all organ systems (Figure 1) [6]. 

The most pervasive long-term effects specific to childhood brain tumors are neurological, neuropsychological, and cognitive dysfunctions [1,7,8]. Cognitive and neuropsychological problems are more frequent in this population compared to survivors of all other pediatric malignancies [3]. It can happen not only because of the experience of a potentially lethal condition and its therapeutic pathway but also due to the involvement of specific cerebral areas, which hits the CNS in a phase of maximum development. Childhood is the time of life during which most CNS development happens; thus, it is not unexpected that CNS tumor survivors commonly experience neurocognitive impairment. A spectrum of disorders ranging from deficits in school functioning to severe dysfunctions can lead to a variable grade of disability, with limitations on everyday activities [3,9]. 

Many patients experience both significant global deficits with decreased intelligence quotients (IQ) and specific neuropsychological deficits such as impaired executive functioning, memory, and processing speed [10]. It is estimated that a large proportion of patients treated for a CNS tumor, ranging from 40% to 100%, will experience a deficit in a neurocognitive domain [11,12]. The simultaneous presence of neurocognitive deficits and neurologic impairments can ultimately lead to a decrease in physical and social functioning and, consequently, to a worsening of patients’ quality of survival [12]. 

In this review, we first summarize the literature on neurologic, cognitive, and neuropsychological outcomes in children treated for brain tumors. We highlight what is known about the complex relationship between biological and treatment factors that influence functional outcomes in this population of patients, focusing on risk factors known to disrupt the normal neurodevelopment pathway. Next, we review current knowledge of monitoring tools and possible strategies to minimize these adverse outcomes. 

We conducted a literature search focusing on neurologic, cognitive, and neuropsychological outcomes of children and adolescents with CNS tumors. The articles were searched in October 2022 in the MEDLINE database (PubMed), combining the search terms as follows: ((child) OR (child, preschool) OR (infant) OR (adolescents)) AND (central nervous system neoplasms/complications) AND ((neurocognitive disorders) OR (neurologic manifestations/complications)). These terms were then differently combined (AND) with (radiotherapy), (chemotherapy), and (proton therapy). The bibliographies of the papers retrieved were analyzed with the aim of identifying records that could have been missed in the primary research phase. Two of the authors examined the titles and abstracts independently. ll retrieved records were examined to include papers meeting the following inclusion criteria: reviews and meta-analysis, research studies, and case series in which the population included patients treated for a brain tumor by neurosurgery, and/or radiotherapy, and/or chemotherapy in childhood or adolescence. Only papers that were written in the English language and those published from the year 2002 up to October 2022 were selected. 

Records for which full texts were not available were excluded; case reports were excluded too. Other exclusion criteria used for the screening of potentially relevant papers were: studies on populations of patients with the diagnosis of a brain tumor in the adult age, studies mainly focused on psychological and emotional outcomes, and studies of in vitro or animal models.

The literature search initially retrieved 320 papers; after the screening of titles and abstracts, 115 records were excluded. The screening of full-text articles assessed for eligibility excluded 119 papers, leaving 86 articles for the final analysis. Figure 2 summarizes the selection process of relevant articles. All full texts of potentially relevant papers were reviewed with a particular focus on biological and treatment risk factors (type of tumor, neurosurgery, chemo/radiotherapy, and patients’ characteristics) and their correlation with cognitive and neurological morbidity. 

Finally, papers focusing on neurodevelopment were included in the discussion. 

## 2. Epidemiology of Brain Tumors in Children 

Primary CNS tumors represent approximately 25% of pediatric cancers, being the most frequent type of solid tumor in children, and are the main non-traumatic cause of death and disability in the first twenty years of life. The annual average age-specific incidence rate of all malignant and non-malignant CNS neoplasms in childhood and adolescence (0–19 years of age) was about 6.23/100.000 from 2014 to 2018 in the United States [4].

The spectrum of brain tumors in childhood is different in terms of location, histology, and prognosis compared to that in adults, suggesting that the pathogenic events are different [13]. During the first two years of life, supratentorial tumors predominate, whereas infratentorial lesions are more common through the rest of the first decade of life; supratentorial tumors again predominate in late adolescence and adulthood [2]. The most common histology in the younger age, between 0 and 9 years, include gliomas and embryonal tumors; in children of age 10–14 years, gliomas and hypophysis tumors are the most common tumor types. In adolescents between 15 and 19 years of age, pituitary tumors and gliomas again predominate [4,13].

It is important to remember that CNS tumor classification has long been based on histological morphological characteristics supported by ancillary tissue-based tests (e.g., immunohistochemical, ultrastructural). The 2016 classification introduced molecular markers as key aspects of classification for a relatively small set of entities. Given the large increase in knowledge of the molecular basis of these tumors, the current fifth edition refers to numerous molecular changes that are crucial for the accuracy of the classification of CNS neoplasms [14]. 

A significant and increasing volume of literature now reports the several long-term effects of brain tumors and their treatment, showing how survival does not come without costs [15]. Pediatric patients with CNS tumors are at high risk of developing physical, neurocognitive, and psychosocial late effects due to tumor site and multimodal therapy combining neurosurgery, chemotherapy, and radiotherapy [15].

## 3. Neurologic and Sensory Late Effects

Children with brain tumors have malignancies and receive therapy that directly affect the brain, and neurological sequelae are common. Although the majority of brain tumor patients present with acute neurologic deficits that will resolve following treatment, some patients experience persistent deficits [3]. Multiple studies have assessed long-term neurologic and neurosensory deficits [9,16]. Survivors of childhood brain tumors may suffer debilitating neurologic impairment, with pain, seizures, and sensory loss among the most predominant. The Children Cancer Survivor Study (CCSS) includes a large retrospective cohort of children and adolescents treated for cancer between 1970 and 1986; data collected on adult survivors of CNS malignancies in childhood show that late neurologic conditions are common in this population and increase with time well beyond the 5-year time point. In the CCSS cohort, a cross-sectional study evaluating 5-year survivors of CNS neoplasms demonstrated an increased incidence, compared to siblings, of early- and late-onset neurologic and neurosensory deficits [9]. Seizures were identified in 25% of patients, more frequently associated with supratentorial tumors [9]. Among the population of survivors without previous seizures in the first 5 years after diagnosis, 33% would experience new onset seizures beyond the 5-year period. Similarly, a high percentage of patients would report visual deficits, with a lower percentage of about 3% for cataracts to a higher proportion of patients complaining of diplopia (17%) [16]. Phillips et al. suggest that pediatric CNS cancer survivors experience additional neurocognitive risk if they develop a seizure diagnosis, with seizure resolution associated with improved attention and memory [17]. In a large cohort of adults survivors of pediatric CNS tumors treated at St. Jude Children’s Research Hospital, the survivors with a history of seizures were at risk of severely impaired academics, attention, and memory dysfunction compared with those without a history of seizures, even after adjustment for cranial radiotherapy (CRT) exposure [18].

Hearing loss is of particular importance in that the cochlea is highly sensitive to toxicity from radiation and platinum-based chemotherapy; sensorineural hearing loss is associated with worse cognitive and functional outcomes, and intact hearing is critical for language development [19,20] Twelve percent of patients in the CCSS cohort report hearing impairment with a statistically significant relationship to posterior fossa irradiation greater than 50 Gy [9]. Ototoxicity is most commonly a result of treatment with platinum analogs that cause direct cochlear damage, possibly due to reactive oxygen species-induced cellular destruction [21,22]. Risk of ototoxicity is compounded with cranial radiation therapy treatment, with doses above 32 Gy increasing the risk of ototoxicity in a dose-dependent manner [23,24]. More recent studies [23] prospectively examined hearing impairment and associated risk factors and found that RT is independently associated with neurosensorial hearing loss even when ototoxic chemotherapy is not administered; Bass et al. also reported children younger than 3 years at RT initiation, with CSF shunt and receiving higher doses to be at higher risk [23]. Brain tumors can also alter the normal neuroanatomical structures of the visual system leading to visual impairment and dysfunction. Visual impairment in childhood is associated with lifelong effects on self-perception, childhood development, and quality of life [25]. Peripheral neuropathy is a common side effect of platinum analogs and vinca alkaloids; these neuropathic symptoms usually resolve following treatment, but, in some cases, they persist for years after therapy [26].

The CCSS cohort demonstrates cumulative incidence of seizures, motor impairment, and hearing loss increases from 5 to 30 years post-diagnosis [27]. Pediatric CNS cancer survivors, unfortunately, are also at increased risk for stroke, which drastically worsens after exposure to CRT [1].

## 4. Cognitive and Neuropsychological Outcomes

Cancer-related cognitive dysfunction affects about one-third of childhood cancer survivors in the US [28]. Neurocognitive late effects in childhood brain tumor survivors are relatively common and play a significant role in modifying Health-related Quality of Life (HR-QOL). Traditionally, the measurement of neurocognitive function has been conducted through the assessment of intellectual quotient (IQ). Declines in IQ are evident in pediatric CNS tumor survivors as early as the first year following diagnosis and treatment, with potential progression over the next 5 to 7 years [29]. Children treated for medulloblastoma are reported to have a progressive decrease in their IQ over time, losing 2–4 points per year over 7 years from the time of diagnosis [30,31]. However, more recently, researchers have focused on neuropsychological functions and have identified the ones at greatest risk, believed to represent core deficits that contribute to broader difficulties [3,29,30]. The most frequently impaired functions include attention, working memory, and processing speed [3,30,32]. 

Executive functions (EFs) deficits are common and are primarily associated with cerebellum-cerebral pathway dysfunction. 

Speech and language deficits are documented among nearly all survivors after post-surgery posterior fossa syndrome [15,32]. 

For some authors, all these functional impairments are thought to be the mechanism behind the decline in IQ. It is reasonable to hypothesize that the neurocognitive decline seen in CNS tumor survivors may be due not predominantly to loss of learned information but rather to a failure in achieving age-related gains in cognitive function [3].

### 4.1. Attention, Working Memory, and Processing Speed

Processing speed is commonly defined as the promptness in performing relatively automatic mental tasks. Some authors report this domain as the first to be impaired after treatment and to be specifically correlated with craniospinal irradiation (CSI) dose [30]. Brinkman et al., however, in a large cohort of adult survivors of pediatric CNS tumors, report a consistent group of 37% of survivors without CRT exposure but severely impaired on at least one measure of processing speed [18]. Given the importance of myelin in the conduction speed of neural impulses, it is not surprising that deficit in this domain is frequent in children with cancer involving the CNS [18]. 

Working memory is a temporary workspace in which information is stored and handled for a brief period of time; working memory tasks have been shown to activate the prefrontal cortex [32]. Several studies have reported specific deficits in working memory in the population of children treated for posterior fossa tumors. In particular, a working memory deficit can emerge very early with a progressive decline over time [30]. Edelstein et al. showed that working memory was the only domain with a continuous decline over time despite a stable IQ score 20–40 years after diagnosis of posterior fossa tumors [33]. 

Attention is the ability to remain alert or focused. It is not a unitary construct, and the subset of skills that falls under the concept of attention includes sustained, focused, selective, divided, alternating, shifting, and resistance to distraction. Attention is demanded by many different tasks, and conclusions about attention functioning can be drawn using subtests of various test batteries [34]. Attention deficits are commonly found in pediatric patients treated for CNS tumors and are reported to appear later than processing speed decline [30]. However, attentional and mnemonic deficits are reported to be present early at diagnosis by some authors [35]. 

These neuropsychological functions are essential for competence and knowledge acquisition, and their dysfunction is considered to be the core deficit explaining academic underachievement [32].

### 4.2. Executive Functions

EFs are different but closely interacting skills required for efficient and appropriate behavior. They are often identified as the ability to control, organize and manage cognitive, emotional, and behavioral responses. These functions rely on the integrity of multiple cooperating brain networks [36]. EFs involve inhibition, mental flexibility, planning and decision-making, abstract reasoning, concept formation, problem-solving, and awareness [37]. There has been some debate about whether it is appropriate to assess executive functioning in children, given that the prefrontal cortex is the last cortical area to reach maturation. In contrast, current research has provided evidence for the development of EF in children and for deficits in these areas following brain injury related to normal development. Therefore, it is important to include measures of EF when evaluating children with cancer involving CNS. Studies on EFs in pediatric CNS tumors have reported these patients to be particularly susceptible to EFs impairments, both in the short and long term. Pediatric cerebellar tumor survivors exhibit similar patterns of impairment in executive functions, in particular in forward-thinking, mental flexibility, and inhibition [38]. 

Consequences of EFs are documented in both low- and high-grade cerebellar tumor survivors [38]. 

Deficits in this area can manifest in daily life as problems with organization, time management, and emotional and behavioral control and will also negatively impact social development [37]. Optimal executive functioning is reported to play a crucial role in long-term functional outcomes. In the context of intact global intellectual functioning, communication, and memory skills, impairments in EFs can cause the greatest handicap for social attainment and adaptive functioning [36] (Table 1). 

## 5. Factors Influencing Neurologic and Cognitive Outcomes

Increased focus on neurocognitive outcomes allowed the identification of important disease and treatment risk factors, and this explains that the survivor’s neurocognitive trajectory is determined by multiple direct and indirect disease- and treatment-related effects (Figure 2) [20,49]. It is crucial to be aware that isolating the role of each variable among the multiple confounding and interacting ones is a major challenge in late effects research [29]. With this caveat, factors with relationships to outcomes include tumors related factors, treatment protocols, and potential moderating variables related to individual patient characteristics and environmental factors (Figure 3) [29]. 

### 5.1. Individual Patient Characteristics 

Younger and female patients are reported to be at greater risk for neurocognitive impairment [3,49]. Increased risk associated with younger age is likely related to the impact of the disease and its treatment on the growth of neural networks [20]. Literature suggests that increased risk for females could be the consequence of the higher susceptibility to treatments affecting the CNS, especially radiotherapy. Irradiation mostly affects white matter, which is considered to be less represented in female than male brains, thus diminishing intact white matter in females and leading to greater cognitive impairment [50]. However, a recent study on survivors of posterior fossa tumors treated with surgery, chemotherapy, and radiotherapy, demonstrated that trajectories of neurocognitive functioning differed between males and females. Females’ cognitive scores are reported to improve in time, on average, compared with males’ scores which deteriorate at 4 years post-treatment [51].

Data reported on a large cohort of children younger than 3 years with intracranial tumors demonstrates that CSI and hemispheric location are associated with significant IQ decline over time [52]. 

Pre-existing developmental disorders and medical comorbidities (e.g., NF1 syndrome) are also risk factors for neurocognitive impairment. 

Increasing evidence exists suggesting that genetic predispositions can influence the effect of cancer and its therapy on neurocognitive outcomes. Individual factors that may modify the vulnerability for treatment-related neurotoxicity and the heterogeneity in cognitive outcomes of this group of patients are not completely known. A new area of research focuses on the study of inherited polymorphisms in genes associated with neural, vascular, myelin, and DNA repair [53]. Variants in the COMT (Catecohol-O-methyltransferase) gene have been associated with cognitive functions in the population of patients with brain tumors. COMT is an enzyme involved in the degradation of catecholamine neurotransmitters, synaptic plasticity, and regulation of dopamine in the frontal lobes, all of which are associated with attention and executive abilities [54]. 

The single nucleotide polymorphism (SNP) rs4680 (G>A) is characterized by a substitution of valine with methionine at codon 158 of chromosome 22q11. Some studies have demonstrated worse cognitive performance in carriers of the Val (G) allele, which leads to higher enzymatic activity and accelerated dopamine degradation with consequent dopamine depletion in the prefrontal cortex. It is possible that in carriers of the variant alleles of rs4680, the disease and its treatment further reduce dopamine availability and consequently compromise the integrity of cognitive functions mediated by the frontal lobes [53]. Neurotrophic factors, including Brain-derived neurotrophic factor (BDNF), are proteins found in central and peripheral nervous systems, which contribute to the growth and integrity of neurons and synapses. BDNF is involved in neural repair and plasticity, and long-term strengthening in the hippocampus. The SNP rs6265 (G>A) is a valine/methionine polymorphism at codon 66 (Val66Met). The Met allele modifies BDNF trafficking and reduces its secretion, resulting in less effective neuroplasticity [53,54].

Polymorphisms in the folate pathway are also involved in increasing the risk of problems in attention and executive functions. Polymorphisms in this pathway may lead to biochemical alterations such as reduced folate levels and elevated homocysteine levels. Since folate is essential for CNS development and function, it is possible that the presence of certain SNPs would be related to neurocognitive deficits after, for example, chemotherapy with methotrexate [55]. Methylenetetrahydrofolate reductase (MTHFR) is an enzyme that catalyzes the production of folate, and the reduction in its activity leads to decreased folate availability. The SNPs MTHFR677 (C>T) and MTHFR1298 (A>C) are the two polymorphisms reported in the literature that are associated with neurocognitive outcomes [55].

Recently other SNPs associated with proteins known to be involved in the neuroinflammatory response and detoxification from reactive oxygen species produced following chemotherapy and radiotherapy have been explored in the literature. Oyefiade et al., with their study on SNPs associated with intellectual outcome differences in medulloblastoma survivors, showed an important role of PPARs (peroxisome proliferator-activated receptors) in suppressing the pro-inflammatory genes following brain injury and, therefore, in modulating the effects of treatment on intellectual outcome [56]. Already in 2009, Barhamani et al. hypothesized a role of deletions in the glutathione-S-transferase (GST) gene in determining worse global, verbal and visuo-spatial IQ in patients treated for medulloblastoma and this has more recently been confirmed [56,57].

### 5.2. Tumor-Related Factors 

CNS tumor diagnosis alone increases the risk of neurocognitive impairment, and this is demonstrated by the fact that before starting treatment, 20% to 50% of patients exhibit cognitive impairment [35]. 

#### 5.2.1. Tumor Location

The majority of studies have focused on post-treatment cognitive deficits in children with brain tumors also because pre-surgery assessment is difficult to perform. A limited number of studies, however, have shown that specific functional deficits in children with brain tumors can be detected before any treatment, with functions like memory and attention being most affected [35]. The severity of hydrocephalus, brainstem infiltration, and the histology of the tumor (medulloblastoma worse than low-grade glioma) is the most significant factors related to the presence of functional deficits prior to neurosurgery [58].

This is a key factor that can explain selective site-dependent deficits, with some tumor locations considered to be at greater neurodevelopmental risk than others. A worse cognitive impairment is evidenced, according to some authors, in supratentorial tumors than in infratentorial tumors, even without performing whole-brain irradiation [59]. Furthermore, left hemispheric lesions are more frequently associated with verbal or language-based deficits, while lesions in the right hemisphere lead to visual perceptual deficits. Other authors report that larger tumor size and infratentorial tumor location are associated with worse neurocognitive outcomes [35,49]. The anatomical location of posterior fossa tumors puts the cerebellar pathway at risk during surgery, and nowadays, it is accepted evidence that a lesion in the cerebellum represents a risk of neurocognitive deficits [30]. 

#### 5.2.2. Obstructive Hydrocephalus 

Additionally, hydrocephalus contributes to determining long-term outcomes for childhood brain tumor survivors. The level of intracranial pressure and the persistence of the pressure on white matter fibers and frontal lobes are considered the main factors influencing mental deterioration in children presenting with hydrocephalus. [60]. Hydrocephalus and shunt placement and revisions are associated with neurocognitive impairment, including lower intelligence, nonverbal reasoning, visual motor integration, memory, and academic skills [12,61]. A history of hydrocephalus requiring shunt placement is reported to confer a 40% increased risk of memory impairment, and this is consistent with known hippocampal vulnerability to hydrocephalus [18].

In contrast, other authors found no relationship between hydrocephalus (with or without shunt placement) and neuropsychological functioning in pediatric LGGs [44].

### 5.3. Treatment-Related Factors 

Treatment for pediatric brain tumors may require neurosurgery, cranial or craniospinal radiotherapy, and/or chemotherapy differently combined, with treatment planning based on patient age and tumor characteristics (histology, stage, location) [20].

#### 5.3.1. Neurosurgery

Treatment of brain tumors with neurosurgery alone is associated with neurocognitive impairment with effects influenced by tumor location and surgical complications. Recent studies suggest that initial cognitive injury starts early in the treatment pathway, possibly pre-radiotherapy, and then persists or worsens during survivorship [20,62]. 

A recent prospective study conducted to assess neuropsychological changes after neurosurgery as the only treatment strategy in pediatric low-grade glioma shows that post-surgery deficits appear early and persist up to 6 years post-surgery. These data also suggest that larger tumor size and supratentorial location are associated with worse neurocognitive outcomes [44]. Risk increases with brain tumors affecting critical brain structures, for example, the hypothalamic-pituitary-adrenal axis or cranial nerves [63]. Supratentorial LGGs located along the midline optic pathway and thalamic tracts may require multiple surgeries and are associated with an increased risk of cognitive morbidity [20]. 

Recent studies [44,64] suggest that children undergoing surgery alone for low-grade glioma without any neoadjuvant or adjuvant therapy remain vulnerable to memory difficulties and executive and motor functioning deficits that persist over time. In contrast to patients treated for CNS tumors requiring RT, those only treated with neurosurgery appear to have an initial injury with stable functioning over time [44]. 

Neurosurgical complications can increase the risk of neurocognitive impairment [8]. Recent studies show that, for example, posterior fossa syndrome, a post-surgical complication occurring in about 30% of patients with infratentorial tumors, is independently associated with a worse cognitive outcome [65]. Although the speech and neurologic sequelae often improve with time and rehabilitation, the literature reports a worse overall neurocognitive outcome for patients who experienced cerebellar mutism syndrome, likely due to damage to white matter tracts in the cerebellar-thalamic-cerebral pathway [3,29]. In particular, survivors experience long-term deficits in cognitive domains, with attention and working memory skills particularly vulnerable to decline over time. Other perioperative factors, such as infections, peritumoral infarctions, and repeated surgeries, have been shown to further compromise intellectual development and social outcomes [8].

#### 5.3.2. Radiotherapy

CRT is often considered the most significant treatment-related risk factor for developing late neurocognitive effects [1,29,66]. Radiation therapy can be responsible for CNS acute, subacute, and late effects, which are described to be progressive in time [67]. The subacute effects can become evident 2 to 6 months after radiation treatment and include radiation somnolence syndrome (RSS), Lhermitte sign, and radiation necrosis [68]. Despite improvements in treatment approaches, CRT directly damages white and grey matter by causing inflammation, angiogenesis, and cell death. Depending on which functional brain structures are irradiated, children who are treated for CNS malignancy often experience substantial long-term impairment in cognitive functions [66,67]. There are different studies in the literature reporting the impact of RT on global intellectual functioning and on specific domains such as memory, attention, and processing speed [1,11,18,69,70]. In cohorts of children treated with conventional radiotherapy techniques, an IQ decline of 2–8 points per year is reported [20]. 

Higher radiation doses and larger irradiated brain volumes are associated with worse outcomes [19,20,71,72]. Large CRT fields are associated with greater neurocognitive impairment, with CSI reported as associated with severe impairment across all cognitive domains [18]. Reductions in boost dose volumes to the tumor bed have resulted in improved neurocognitive outcomes, and reducing the dose to sensitive brain regions have demonstrated better neurocognitive outcome in patients treated for medulloblastoma [49,69]. In medulloblastoma patients, the post-RT intelligence quotients were reported to be 10–15 points higher after a whole brain dose of 23.4 Gy vs. 36 Gy, but other studies demonstrated a dose response in the 18–36-Gy range. Differences in results between the studies can be explained by the failure of small samples to overcome the complex interactions among dose, volume, patient age, and follow-up length [73].

Exposure to CRT at younger ages is associated with worse outcomes because the proliferation of neural precursor cells is more active shortly after birth and declines with age [49,66]. Cranial radiation also affects the growth of new neurons in the hippocampus leading to the decreased hippocampal volume associated with specific memory deficits [29].

Neurocognitive dysfunction progresses with time since CRT and brain imaging demonstrates a decline in white matter integrity. The effects of radiotherapy can be detected for at least 50 years after exposure and can be explained by the persistent impact on proliferating oligodendrocytes (myelin) and/or precursors of other cell types [66]. 

Indeed, the burden of cognitive outcomes among infants and toddlers post-CRT in the past few decades led to changes in treatment plans to delay or eliminate RT in children younger than 3–5 years of age. Current data do not define an age threshold beyond which cranial irradiation is no longer a risk factor for cognitive impairment. The nonlinear development of specific cognitive abilities in childhood and adolescence suggests that radiotherapy may be more detrimental to specific abilities than others in certain age ranges [20]. For these reasons, cognitive and neuropsychological outcomes have been included as endpoints in recent trials with the aim of reducing or eliminating radiotherapy, when possible, without affecting prognosis. Headstart trials focused on intensive chemotherapy followed by hematopoietic stem cell rescue with the goal of avoiding or postponing radiotherapy; results published from Headstart II and III showed stable neuro-cognitive functioning over time, suggesting this approach to be effective in sparing CNS development in young children [74].

Advanced CRT techniques (i.e., intensity-modulated radiotherapy, proton therapy) have improved the precision of dose delivery, resulting in significant reductions in doses given to healthy tissues; for example, proton CRT is expected to provide similar disease control while minimizing dose to healthy tissue but outcome studies are just emerging [75]. This increased conformity of modern radiotherapy techniques could be the reason why the results of more recent published studies show better outcomes compared to previous studies on cohorts treated decades ago [47].

Early proton findings are encouraging, particularly in patients treated with focal proton therapy, who have shown stable and preserved cognitive functioning [20]. In some recent studies, proton therapy has been shown to less negatively impact neurocognition (relatively intact global intellectual functioning, attention, executive functions, and only impaired processing speed) [76]. Gross et al. described a cohort of children treated at a mean age of 7.4 years, in which proton therapy is associated with favorable outcomes for processing speed and intelligence [77]. However, this potential benefit of proton therapy vs. photon radiotherapy may disappear when patients undergo craniospinal irradiation [76]. A recent prospective longitudinal trial assessed neurocognitive outcomes in pediatric patients treated with proton CSI or focal irradiation versus surgery alone; authors showed stable neurocognitive scores over 6 years for patients treated with focal proton therapy or surgery only compared to significate decline in patients who underwent proton CSI [64]. Overall, early studies on proton therapy are encouraging, mainly for those patients treated with focal therapy who have shown stable intellectual functioning even in some traditionally radiosensitive domains [64,78].

#### 5.3.3. Chemotherapy

Effects of chemotherapy alone are difficult to isolate in the context of other multimodal treatments such as surgery and CRT, as well as in the presence of other tumor-related factors, but it has been demonstrated that chemotherapy also plays a role in neurocognitive decline [3,12,79]. Neurotoxicity caused by chemotherapeutic agents is a frequently observed adverse effect, and both peripheral and central neurotoxicity can occur, with a potential injury of chemotherapy on neural progenitor cells and healthy brain tissue. Specific chemotherapy agents are known to represent a direct risk for neurocognitive impairment as well as an indirect risk related to ototoxicity [29]. Central neurotoxicity can range from minor cognitive deficits to severe encephalopathy with dementia and occurs more commonly with agents such as methotrexate, cytarabine, and ifosfamide. Much of the knowledge on chemotherapy-related neurocognitive dysfunction has been obtained from studies of pediatric patients with leukemia, and methotrexate is one specific drug that has been linked to neurocognitive deficits. Such negative effects from chemotherapy are thought to be related to injury to early nerve progenitor cells, hippocampal toxicity, and oxidative stress [1].

### 5.4. Environmental Factors

Environmental factors, including low socioeconomic status and high stress levels, may increase the risk of poor neurocognitive and psychosocial outcomes. Prior studies have shown an association between parental stress/distress, socioeconomic status, educational support, and the child’s coping ability, potentially leading to maladaptive responses that can affect the entire course of treatment [80]. Pre-morbid education, income, availability of family support, and access to health care, including neuropsychology, are challenging to examine but may contribute to a person-centered approach to understanding cognitive risk.

## 6. Neurodevelopmental Framework

Research in the last two decades has shown how multiple factors contribute and interact to determine neurologic impairment and cognitive risk in this population. Survivors have been shown to remain at risk for progressive neurologic and cognitive decline for years after diagnosis. What we need to consider is that all these risk factors impart significant injury to a developing brain, which has some peculiar characteristics that can explain the detrimental effect of the tumor and its treatment in this specific population.

The human nervous system is never static, and development occurs across the lifespan. However, the rapidity of development in the period of childhood and adolescence calls for a specific developmental focus when conducting assessments in this age group [34]. It is important to understand the dynamic nature of the child’s nervous system in order to adequately understand how development may be interrupted or changed by an illness, injury, or toxic exposure.

Childhood is the period during which most brain development occurs and represents a period of incomparable brain sensitivity [3]. At birth, infants have more than 100 billion neurons [81]. In the first two years of life, the brain undergoes a period called transient exuberance, when new synapses are established between neurons. These connections are modified through a process called pruning, which is a rapid elimination of synapses that peaks in adolescents and differs between brain regions. In this process, frequently used synaptic connections are strengthened while unused connections are pruned; this is why there are critical periods during development when the brain expects certain experiences from the environment in order to develop [82]. The neural circuits are characterized by patterns of connectivity that are mainly established by the genetic blueprint and are consolidated by the presence of a preferred stimulus, so the absence of that stimulus potentially leads to permanently abnormal connections. On the other hand, the neural circuits are highly “motivated to change,” and the genetically encoded multiple potential connections are selected and can commit to one pattern or the other based on the stimulus [82]. Additionally, myelination is an important neurodevelopment process that begins in the prenatal period and continues into childhood and adult years. The dynamic nature of the child’s nervous system explains the concept of plasticity, which is the capacity of the brain to continuously change its structure and, ultimately, its function.

Diagnoses and treatment of pediatric and adolescent cancer coincide, therefore, with periods of substantial neurodevelopmental change and can potentially disrupt the gradual emergence of functional neural connections, shift the developmental cascade, and have neuropsychological effects in subsequent developmental stages [83].

The impact of pediatric cancer as an adverse childhood experience needs to also be considered when evaluating psychological and neurodevelopmental outcomes. Childhood adversity is defined as an experience that is likely to require neurobiological adaptation by the average child, and that represents a divergence from the expectable rearing environment. Some authors [84] emphasize the role of childhood adversity, which has not been considered for years in the prior existing literature despite more recent research showing that exposure to more commonly studied adverse childhood experiences strongly imprints on neural development. Although other adverse childhood experiences (e.g., violence, abuse) are different in many aspects from the experience of childhood cancer, they do share the common element of exposure to a threat to life or physical wellbeing. Marusak et al., therefore, claim that the double hit of early threat and cancer treatments likely alters neural development and, consequently, cognitive, behavioral, and emotional outcomes [84].

## 7. Strategy of Monitoring and Potential Intervention

Neurocognitive injury in pediatric brain tumor patients is a long-term toxicity that can begin early at the onset of disease and persist even lifelong, with the necessity to be monitored closely in children with a history of any brain tumor.

A better understanding of which patients are at major risk and how to organize timely screening and follow-up still remain questions under investigation.

### 7.1. Who Is at Major Risk?

Each child or adolescent with a diagnosis of a brain tumor is at risk for immediate and then persistent or progressive neurocognitive impairment.

Younger children with infratentorial tumors are considered to be at increased risk [67].

Children frequently have posterior fossa tumors if compared to adults; the effect of these lesions on cerebellar structures has a significant impact on neurobehavioural outcomes, given the cerebellum’s role in cognitive and executive functions. The most common malignant CNS neoplasm of childhood is medulloblastoma, which typically arises in the posterior fossa and requires aggressive multimodal therapy, including surgery, chemotherapy, and radiotherapy. Cognitive impairment of children treated for medulloblastoma is primarily ascribed to the impact of radiation on white matter development [32,44]. Neurocognitive dysfunction progresses over time, especially in patients treated with radiotherapy, and brain imaging shows a decline in white matter integrity with increasing age after CRT [49].

Different from what one might think, given the substantial benign nature of low-grade tumors, pediatric patients undergoing resection only without adjuvant therapy are also at risk for neurocognitive deficits. In contrast to children receiving radiotherapy, these patients tend to have an initial insult with stable functioning over the years [44].

### 7.2. How to Screen and Follow-Up with These Patients?

The neuropsychological assessment bridges the medical and biopsychosocial model of care and helps clinicians identify possible functional deficits, allowing them to objectively classify health-related domains within the World Health Organization’s International Classification of Functioning, Disability, and Health (ICF) [85].

Neuropsychological assessments are particularly resource intensive, with a single assessment estimated to take up many hours of clinician and children’s time. It is, therefore, essential to understand the risk profile of children who may require neuropsychological assessment in order to accurately select appropriate neuropsychological test batteries [86]. The heterogeneity in assessment batteries and the significant burden in terms of costs and time have been reported to be important limitations to the evaluation of cognitive impact in trials in children [28].

Additional issues exist in selecting appropriate tools of evaluation specific to age, using both direct and indirect measures.

Therefore, the Brain Tumor Quality of Survival Group of the European Society of Pediatric Oncology (SIOPE) has proposed a “core plus” approach, identifying a small core battery of direct cognitive assessments to be completed with supplementary measures in specific cases. They suggest using widely available tests on a global scale (e.g., versions of the Wechsler Intelligence Scales for Children) for the core battery of direct assessment.

A detailed analysis of direct and indirect measures of cognitive functioning is beyond the scope of this review (for a detailed battery of neuropsychological tests for children below 5 years and 5 years and over, see, respectively, [87] and [88]).

Comprehensive neuropsychological assessment and neurological reports reveal more and more diverse problems than could be inferred from the IQ measurements alone, which is no more sufficient in this complex interaction [89].

For children with brain tumors, the complexity of care throughout treatment and survivorship requires a multidisciplinary approach. This means that cognitive and neuropsychological evaluation needs to be integrated with a detailed assessment of neurologic and neurosensorial impairments. Neurological and sensory abnormalities detected during a careful clinical examination should be considered major risk factors for subsequent cognitive impairment [32].

Vision impairment can also impact performance on some neuropsychological tests and needs to be considered as a risk factor before and after RT. Achieving seizure resolution is a crucial step toward improving neurocognitive outcomes in CNS cancer survivors [17].

Most importantly, cognitive and neuropsychological report, together with a detailed assessment of medical comorbidity, generates individualized recommendations for how these problems may be addressed, in particular in the educational and rehabilitative context [86].

Together with clinical evaluations, the development of biomarkers of neurotoxicity can facilitate early identification of impending neurocognitive dysfunction [28]. Biomarkers of cognitive impairment are emerging in the field of neuroimaging, with the use of advanced techniques highly informative on functional networks and connectivity, such as diffusion tensor magnetic resonance imaging (DTI) and functional MRI (fMRI). The majority of studies that examined brain structure or function in child or adolescent cancer survivors reported variation in regional gray matter or cortical thickness, as measured by structural MRI, or white matter macrostructure abnormalities, as measured by diffusion tensor imaging [84]. White matter integrity has been shown to be inferior in patients treated with adjuvant therapy if compared to those not receiving therapy, especially RT or high-dose methotrexate [49].

### 7.3. When to Start Follow-Up and for How Long It Is Recommended to Continue It?

The different patterns of dysfunction, time of onset, and evolution demonstrate the importance of starting early monitoring in this population (regardless of tumor histology and site) and continuing surveillance during and after treatment, ideally lifelong.

The Children’s Oncology Group Long-Term Follow-Up Guidelines for Survivors of Childhood, Adolescent, and Young Adult Cancers offer professionals a summary of potential late effects associated with cancers and their treatment and identify basic recommendations for evaluation and management. For patients with CNS involvement, the guidelines include recommendations for follow-up of late neuropsychological effects [90].

Cognitive and neuropsychological assessment is a sensitive tool for monitoring developing brain systems and brain functions, both acutely and longitudinally, across time. Changes in performance over time can be indicative of emerging late effects or early signs of disease recurrence. Early detection of delayed acquisition or decline of neuropsychological functioning allows earlier interventions that can attenuate subsequent disability. Starting intervention early in time, during treatment, aim to correct aberrant neurodevelopmental trajectories [84].

Therefore, the Children’s Oncology Group’s long-term follow-up guidelines recommend a baseline evaluation followed by periodic neuropsychological evaluation as clinically indicated according to therapeutic exposures. Serial assessment presents a particular challenge in selecting the most appropriate instruments, detecting meaningful changes, and determining when and how often to test. At least, all childhood cancer survivors at risk for neurocognitive impairment should undergo a baseline evaluation at the time of entry into long-term follow-up, even if no overt manifestation of CNS injury is detectable. A re-evaluation is recommended in the phase of school transitioning or when new difficulties emerge or are reported by caregivers [91].

At all ages, there are issues with the optimal timing of baseline assessment, which can potentially be performed at diagnosis, before surgery, before radiotherapy, or even after treatments [87].

Studies of the cognitive sequelae of pediatric CNS tumors have focused almost exclusively on long-term cognitive impairments identified after treatment and not on pre-treatment cognition, even though a baseline evaluation is recommended. One reason could be that some children are too ill or distressed at the time of diagnosis for a complete assessment to be undertaken, or there is not enough time between diagnosis and the first treatment, usually surgery [92]. When possible, according to the clinical and psychological conditions of children, a pre-surgical evaluation is suggested to establish a baseline of neurodevelopment. Some recent studies also underline the importance of a structured neuropsychological evaluation before and after radiotherapy to better evaluate RT-induced neurocognitive decline [89].

A one-time neuropsychological evaluation represents a single picture of development and functioning but does not provide adequate information about the trajectory of neurodevelopment. Completing serial neuropsychological assessments provides more comprehensive information on the onset, stability, and severity of neuropsychological disruptions and is an essential tool in the clinical care of this vulnerable population [85]. In contrast, to other acquired injuries such as traumatic brain injury or congenital developmental disorders, neuropsychological deficits in this population often emerge slowly and can intensify over time, and a one-time evaluation can provide only a single snapshot of the patient and not a complete monitoring of the appearance of deficits in time. This is the reason why the SIOP-E brain tumor group recommends a complete assessment during follow-up at 2 and 5 years from diagnosis and at the age of 18 years as an end time point (Figure 4) [87].

Understanding of neural networks and brain plasticity suggest that early interventions can achieve an improved neurocognitive outcome.

Both pharmacologic and psychological/educational interventions are reported to avoid the progression of neurocognitive dysfunctions [28] (A detailed list of possible interventions is beyond the scope of this review).

## 8. Conclusions

Although survival rates in children diagnosed with brain tumors have significantly improved in the last decades, especially for some tumor types, it is now clear that the disease itself, together with aggressive treatments, put this population at substantial risk for functional deficits in multiple domains, reducing their quality of survival. It is, therefore, increasingly important to understand the pathophysiology and risk factors of late developmental effects.

The neurocognitive assessment in the pediatric population of patients with a CNS tumor is a complex part of the therapeutic path.

From a clinical point of view, cognitive and neuropsychological assessment is strongly recommended, starting with a baseline evaluation at diagnosis when the child’s clinical condition allows it. Indeed, it is necessary to take into consideration the strong psychological distress of this kind of diagnosis for the child and the family.

It is also strongly recommended to extend long-term clinical and neurocognitive follow-up to detect the late effects that manifest years after treatment, with particular attention to developmental trajectories and times of education transitioning.

Early detection of delayed acquisition or decline of neuropsychological functioning allows earlier individualized habilitative/rehabilitative interventions that can reduce subsequent disability and improve HR-QOL.

From a research point of view, understanding which cognitive domains are more susceptible to injury and decline and for what group of children and treatment there is an increased risk could have important clinical implications and guide the conceptualization of new trials. The results of the studies included are limited by methodological problems related to the great heterogeneity of the population, recruitment criteria, and the use of different measures of outcome. Further research is needed to answer the unresolved questions also concerning the neurobiological mechanisms underlying the neuro-developmental and cognitive trajectories of these children.

In addition, longitudinal studies that include multiple outcome measures, both functional and related to multiple domains of neurocognitive development rather than IQ, are needed. Our review examined risk factors for neurocognitive and neuropsychological impairment, a very complex and broad topic in itself. The limitation of this review is that we did not conduct a systematic review of the instruments used to assess neuropsychological outcomes, which are very heterogeneous. Furthermore, the data reported in the studies refer to a wide time span in which treatment modalities have changed substantially. Therefore, neurocognitive outcomes may be different for patients treated today than for the cohort of patients treated in recent decades.

## Figures and Tables

**Figure 1 children-10-00472-f001:**
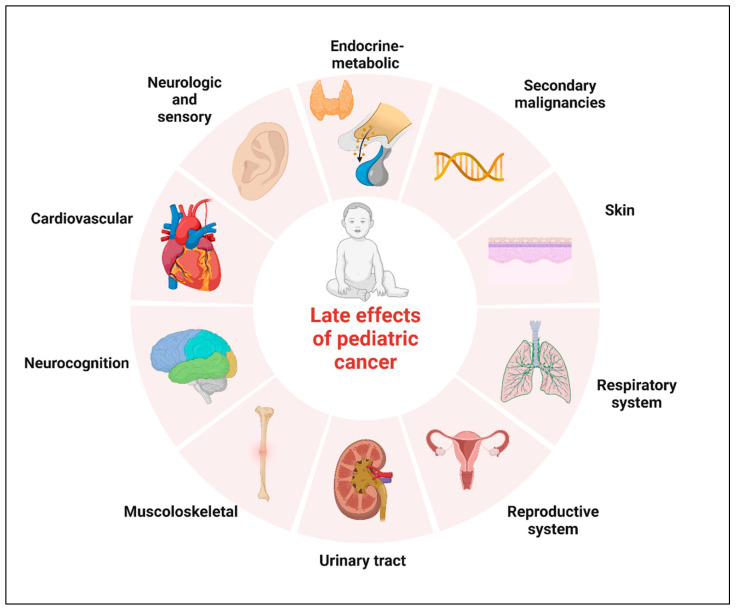
Potential complications of childhood cancer by organ system. Created with BioRender.com.

**Figure 2 children-10-00472-f002:**
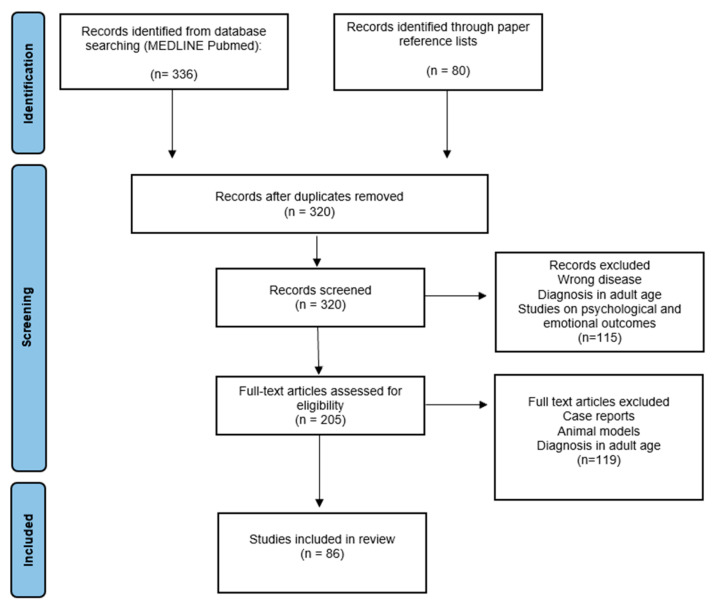
PRISMA flow diagram for selection of articles.

**Figure 3 children-10-00472-f003:**
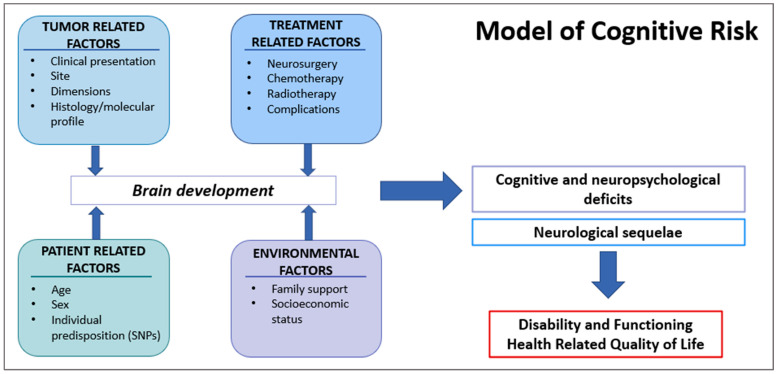
Interaction of multiple factors representing potential injury to the developing brain and leading to neurologic and cognitive/neuropsychological deficits. Ultimately, these late effects together are associated with disability in everyday activities and, therefore, with impaired Health-Related Quality of Life.

**Figure 4 children-10-00472-f004:**
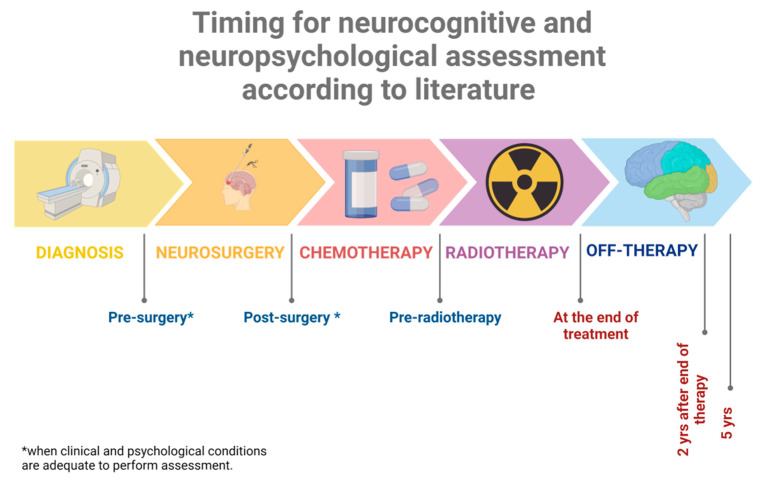
Timing suggested for baseline evaluation and subsequent follow-up assessments. At any time point, during treatment or after the end of therapy, a reassessment is recommended in the phase of transition to new schools or when new difficulties emerge. Created with Biorender.

**Table 1 children-10-00472-t001:** Cognitive domains impacted by tumor and its treatment and studies describing characteristics and risk factors of impairment.

Cognitive Domain	Author, Year	Population	Main Findings
Global intellectual functioning	De Ruiter et al. [38]	Meta-analysis summarizing neurocognitive outcomes of 1082 pediatric brain tumor survivors	PBTS scored on average 0.54SD to 0.90SD lower on the WISC-III scales than the normative sample, with PIQ scores being even more depressed than VIQ scores
	Brinkman et al. [17]	224 adult survivors of CNS pediatric tumors	20–30% of survivors demonstrated impairment on performance-based measures of intellect compared to expected 2% in the general population
	Lafay-Cousin et al. [39]	16 Atypical teratoid/rhabdoid tumor survivors	Overall impaired neurocognitive outcome while treated with a radiation sparing approach
	Clark et al. [40]	43 survivors of focal low-grade brainstem gliomas	Measures of intelligence quotient significantly lower than normative, despite focal disease and treatment targeting subcortical areas
	Moxon-Emre et al. [41]	113 patients treated for medulloblastoma	Patients treated with reduced dose craniospinal irradiation plus tumor bed boost showed stable intellectual trajectories while those treated with higher doses and larger boost experienced decline.
	Roncadin et al. [42]	29 astrocytoma and 29 medulloblastoma survivors	Greater perioperative and short-term medical adversity contributes to lower IQ in the long term
	Margelisch et al. [35]	20 CNS tumor patients compared to 27 control patients (other type of cancer) at diagnosis	Mean IQs of patients with brain tumor lie within the normal range at diagnosis
Executive functions	Law et al. [43]	25 children treated for medulloblastoma with surgery, CRT and chemotherapy and 20 healthy controls	EFs deficits are found children treated for medulloblastoma compared to age-matched peers. Selective deficits in cognitive efficiency, problem-solving and working memory. Specific damage to cerebrocerebellar circuitry.
	Heitzer et al. [44]	32 patients treated for LGG with surgery only	Supratentorial LGG and history of seizures: greater impact on executive functioning
	Koustenis et al. [45]	42 pediatric posterior fossa tumor survivors (mean age 14.63 years	Pediatric cerebellar tumor survivors show similar pattern of impairment in executive functions in particular in forward-thinking, mental flexibility and inhibition
Memory	Margelisch et al. [35]	25 children treated for medulloblastoma with surgery, CRT, and chemotherapy and 20 healthy controls	Memory and attention are the principal domain found to be impaired at diagnosis before treatment
	Decker et al. [46]	29 PBTS treated with chemotherapy and CRT	Associations between hippocampal subfield volumes and short-term verbal memory
Attention	Margelisch et al. [35]	25 children treated for medulloblastoma with surgery, CRT, and chemotherapy and 20 healthy controls	Memory and attention are the principal domain found to be impaired at diagnosis before treatment
Processing speed	Weusthof et al. [47]	103 CNS pediatric patients treated with photon therapy, proton therapy or surgery alone	Processing speed is the most vulnerable domain with decline over time in both photon and surgery cohorts
	King et al. [48]	57 neurotypical controls and 57 survivors of childhood brain tumors	Processing speed appears to be the central cognitive skill that disrupts the other core cognitive skills of attention span and working memory, and all three make a unique contribution to IQ and academic achievement

PBTS: pediatric brain tumor survivors; PIQ: performance intelligence quotient; VIQ: verbal intelligence quotient.

## Data Availability

Not applicable.

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
