# Peer review of "Neuropsychological Outcomes of Children Treated for Brain Tumors"

_children, 2023, doi:10.3390/children10030472_

Round 1

Reviewer 1 Report

Thank you for the opportunity to review your manuscript, Neuropsychological Outcomes of Children Treated for Brain Tumors. How tumor of the central nervous system affect developmental domains and day-to-day functioning are important topics for study.

I think this manuscript has some clear strengths – especially its relevance for detecting delayed acquisition or decline of neuropsychological functioning and for developing interventions to attenuate disabilities. At the same, I have methodological concerns that can be reformulated so I can recommend this study for publication. Below I detail these concerns. I look forward to reading a revision of this article, if the authors choose to revise. 

The article is based on a traditional literature review on brain tumors and its neuropsychological implications. The authors examine research reports regarding epidemiology of brains tumors in children, neurologic and sensory late effects, cognitive and neuropsychological outcomes as well as factors that influence these outcomes and monitoring and potential interventions. However, the literature review it seams subjective, due to the substantial reliance on the author’s pre-exiting knowledge and experience and as it does not present an unbiased, exhaustive and systematic summary of the content. Considering these limitations, I recommend a scoping review or a systematic review as alternatives to clarify the concepts.

The authors already integrate in the article some presuppositions of a systematic review, such as the identification of eligibility criteria, research sources and some research strategies. There are some information missing regarding the procedures to develop a systematic review. I recommend a decision-making process about the method developed and, in case of wanting to proceed with a systematic review, I recommend reading articles related to PRISMA - Preferred Reporting Items for Systematic Review and Meta-Analyses (2020) to rigorously include the necessary procedures for conducting a systematic review.

In the conclusions the authors suggested future studies, but they didn’t mentioned limitations of the literature review. I think they should include limitations, namely if the option is to maintain the traditional literature review.

Author Response

Dear reviewer,

Thank you very much for your kind appreciations and for your comments and questions, which allows us to clarify the aim of our review.

We have tried to explain more in detail the process of articles selection with inclusion and exclusion criteria defined more in detail with the aim of giving a structured approach to the methodology of our . As suggested, we agree that explaining clearly exclusion and inclusion criteria for literature search and establishing relevance criteria of selection, can improve the quality of the narrative review, applying some aspects of the methodological rigor of a systematic review.

As requested, we added limitations of our review together with needs for new researchs in the conclusion

Reviewer 2 Report

None 

Author Response

Dear reviewer,

Thank you very much for your judgements. There are no specific criticisms to reply.

Reviewer 3 Report

I want to thank for the possibility to review this review article regarding neuropsychological outcomes of children treated for brain tumors. The article addresses an important area of research within pediatrics. It is written in a way that it is easy to follow and the authors have created images that illustrate the message well. I have some comments and questions.

The authors present how they searched for a literature and reviewed the articles but they do not present criteria how the included papers were selected and summarized. A  systematic review ( for example according to PRISMA guidelines) would have better. The authors should at least describe on what bases the articles were selected.

The review covers very many different aspects of the topic from epidemiology to risk factors, potential interventions and makes the text at times rather general. One would expect that a review summarizes the previous literature and makes a synthesis of that. Now in many places of the article this synthesis is difficult to find, and the authors list previous results after each other without a synthesis. As an example (at rows 377-384) the authors write first "Treatment with CRT is associated with a 15- to 25 point decline in IQ compared to children who do not receive cranial radiotherapy." and then two sentences later "One systematic review and meta-analysis further reported and average loss of more than 10 points IQ scores in those treated with RT compared to those not irradiated". Also, there is lack of information about a time span since RT these numbers come from. 

The authors repeat some issues several times. An example of such is that childhood is the period when most of the the brain development occurs, that it is sensitive and vulnerable time of development, which the authors take up many times and even several times in a same paragraph. 

In some parts more exactness and directness would be needed. For example paragraph 7.1 who is at major risk does not really give an answer to that question.

Please check the use of abbreviations for example cranial radiotherapy can be shortened to CRT throughout the paper. 

In the introduction the authors state that survival at 5-years approaches 75% and that low-grade gliomas nearly reaches 90% at 20 years. There are articles showing higher survival which could be used as better references. Please mention also if that it is numbers from the developed countries. 

Table 1 presents studies of different domains. It shows for example only two small articles regarding memory, it is unclear at which bases the articles in the table are selected. Please explain.

The authors state at row 129 that 16% of the survivors report new onset blindness. This number is very high and the reference does not seem clearly show that, please check that. 

The paragraph regarding effect of chemotherapeutic agents should be developed. Authors state that Ifosfamide as chemotherapeutic agent commonly associated with central neurotoxicity when the main heading is   neurologic and cognitive outcomes. Although ifosfamide is associated with acute CNS symptoms during infusion, what is it's the role for "outcomes". It would be beneficial if the authors could summarize what is know about the role in chemotherapy in CNS tumor survivors, in addition to that it is difficult to study. 

The authors write a whole paragraph discussing the potential effect of genetic polymorphisms in the folate pathway to attention and executive functions based on one study. It is important to take up that the study is from leukemia patients. There is a recent study regarding  It would be better to focus on the polymorphisms studied specifically in CNS tumors such as PPAR, GST.

Author Response

Dear reviewer

Thank you very much for your kind appreciations and for your comments and questions, which allows us to clarify the aim of our review. We have tried to elucidate the concerns you have detailed.

1)The article is based on a traditional (narrative) literature review and this choice has been driven by the fact that the aim of the paper is to extensively scope the broad topic related to neurological, cognitive and neuropsychological effects of brain tumors in children, rather than focusing on a unique query (which is a characteristic of the systematic review). We decided to use the form of narrative review, because some issues are better addressed with the wider scoping of a narrative review.
The aim of this review was to give a global view of multiple risk factors interacting in the treatment of pediatric brain tumors (individual patient characteristics, tumor related factors, treatment related factors, environmental factors), determining neurocognitive and neuropsychological outcomes. Each risk factor and its role in modifying the outcomes is a very complex topic and we think that maybe a systematic review could be appropriate for addressing each single topic one at a time. Thanks to your suggestions we have understood that maybe it would have been better to clarify this point through a more detailed title, for example: “Neuropsychological outcomes of children treated for brain tumors: risk factors and strategy for monitoring”.

However, we agree with your opinion  that subjectivity in study selection and synthesis is one of the main weaknesses ascribed to narrative reviews.

Therefore, considering your  comments we explain more in detail the process of articles selection with inclusion and exclusion criteria defined more in detail, not with the ambition of writing a systematic review but with the aim of giving a structured approach to the methodology of our narrative review. Based on your criticism , we think that explaining clearly exclusion and inclusion criteria for literature search and establishing relevance criteria of selection, has improved the quality of the narrative review, applying some aspects of the methodological rigor of a systematic review.

2) Thank you for your comment that has led us to make a synthesis in different points of the paper. For some points this has not been possible because the evidence in literature are not enough to summarize the results in a single statement but data are conflicting and need to be listed.

3) We have tried to reduce repeated issues throughout the paper to make the concepts more concise and less redundant.  

4) With the literature available nowadays it is really challenging to give a precise answer to that question; the aim of that paragraph title was to underline the take home message that everybody who has been treated for a brain tumor in childhood or adolescence, even only with neurosurgery without additional therapy, is at risk for cognitive impairment and needs to keep an eye on it. We have underlined what is reported in literature that is a higher risk among younger children with infratentorial tumors, which are more common in childhood.

5) We have checked the abbreviations throughout the paper, thank you for underlying this.

6) Thank you for the clarification. The most recent data from the Central Brain Tumor Registry of the United States, which is the largest population-based registry exclusively on primary CNS tumors, show higher survival than the ones we reported, as you stated. Pilocytic astrocytoma are one of the most commonly diagnosed histology of primary brain tumor in children less than 20 years of age and have very high survival. PAs are clinically considered and classified as grade I tumors and therefore non-malignant as per WHO classification. However for the purpose of cancer registration reports in North America PAs have historically been reported as malignant tumors. This reclassification of PAs, as reported by Ostrom et. al., has a significant effect on the estimates of relative survival after a diagnosis of brain tumor in the population of children 0-19 years old and this can explain some differences in survival data reporting in literature.
According to CBTRUS 5-year relative survival for malignant tumors is 76% and for non-malignant tumors is 97,9%; when PAs are reclassified as non-malignant tumors 5 year relative survival for malignant tumor decrease to 68.9% and survival for non-malignant tumors has no substantial change.  

7) Table 1 has not the aim of been exhausting on studies available for different domains; we have decided to report only papers in which the authors have done a direct neuropsychological evaluation rather than indirect evaluation (for example parents reports). For the analysis of articles reported in the table we have decided to exclude many articles because of the indirect measures used for the evaluation of domains. This is one of the critical points of the available literature on neuropsychological outcomes of this population.

8) Row 129: the sentence was imprecise in reporting the results of the cited studies. We have corrected the statement, thank you for pointing this out.

9) It is very difficult to isolate the role of chemotherapy given the complex interactions of different factor and the fact that in brain tumor chemotherapy is often administered together with radiotherapy. We have detailed more what is the role of chemotherapy in determining not only an acute toxicity but also cognitive late effects.

10) Thank you for the suggestion of the recent study on PPAR and GST specifically studied in CNS tumors. References 51 and 52 (50 and 51 in the first version because of some changing with major revisions) too are studies on different polymorphisms in brain tumor patients, only the actual 53 (previously 52) is a study about leukemia patients but focusing on the folate pathway which can modulate the impact of MTX therapy, which is associated with increased risk of poor cognitive outcome in brain tumor patients too.

v

Round 2

Reviewer 2 Report

I think the revised version of the paper is now more than acceptable.

Even if the paper does not involve major new information for professional, it is still a chance to check in on the point.

Reviewer 3 Report

I think the authors have answered the questions well and made the corrections needed.